# A Systematic Literature Review of the Predictive Maintenance from Transportation Systems Aspect

**Olcay Özge Ersöz, Ali Fırat İnal \*, Adnan Aktepe, Ahmet Kürşad Türker and Süleyman Ersöz**

Department of Industrial Engineering, Kirikkale University, Kirikkale 71450, Turkey
\* Correspondence: afinal@kku.edu.tr; Tel.: +90-535-7141992

**Abstract:** With the rapid progress of network technologies and sensors, monitoring the sensor data such as pressure, temperature, current, vibration and other electrical, mechanical and chemical variables has become much more significant. With the arrival of Big Data and artificial intelligence (AI), sophisticated solutions can be developed to prevent failures and predict the equipment's remaining useful life (RUL). These techniques allow for taking maintenance actions with haste and precision. Accordingly, this study provides a systematic literature review (SLR) of the predictive maintenance (PdM) techniques in transportation systems. The main focus of this study is the literature covering PdM in the motor vehicles' industry in the last 5 years. A total of 52 studies were included in the SLR and examined in detail within the scope of our research questions. We provided a summary on statistical, stochastic and AI approaches for PdM applications and their goals, methods, findings, challenges and opportunities. In addition, this study encourages future research by indicating the areas that have not yet been studied in the PdM literature.

**Keywords:** predictive maintenance; transportation systems; systematic literature review

## 1. Introduction

The aim of the development of technology is to increase productivity in areas such as production, maintenance and quality in enterprises. Factors such as ineffective periods that may occur due to malfunctions in production and defective products affect productivity significantly. A maintenance strategy that is pre-determined and implemented at the right time is an important factor in increasing efficiency.

Maintenance strategies, also called maintenance policies in the literature, include maintenance activities such as the parts' replacement, renewal and repair required to ensure the continuity of the health status of the assets in the enterprise throughout their life and to fulfill the operational functions. Maintenance strategies have been classified in different ways by many researchers. In the literature, four general maintenance strategies are generally mentioned: preventive; predictive; corrective and prescriptive maintenance [1–5]. In Figure 1, a visual summarizing the working techniques of different types of maintenance is given.

PdM is the process of planning maintenance activities and performing maintenance using various forecasting methods for potential failures before the failure occurs. PdM activities use data science to predict when equipment might fail. Based on the data, the fault point is estimated and maintenance activity can be planned before this point. The aim is to provide the sustainability of the system by planning the maintenance process at the most appropriate moment before the life of the equipment expires [6–9].

In the last decade, besides the increase in automation, developments in neural networks and machine learning have also been achieved. With the growth of the stored data and the evolution of GPU-based and similar processors that can process complex algorithms that can work on these data, neural networks consisting of more units and hidden layers have become trainable [10–13].

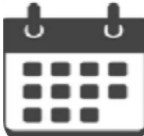

**Preventive Maintenance**

*Maintenance is performed due to a schedule*

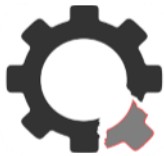

**Run to Failure (R2F) or Corrective Maintenance**

*Maintenance is performed when an equipment failure occurs*

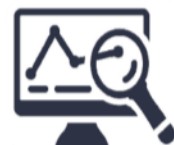

**Predictive Maintenance**

*Monitoring of an equipment using sensors or analytical tools*

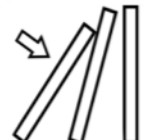

**Prescriptive Maintenance**

*Calculate the effects of the variables causing the failure*

**Figure 1.** Different types of maintenance.

Artificial intelligence and deep concepts fulfill many important purposes in different fields. In particular, with the concept of deep learning, it is ensured that meaningful and important inferences are obtained from large data groups [14–16].

PdM techniques encourage non-damaging testing methods such as acoustic, infrared, sound level, oil analysis, vibration analysis and thermal image recognition that measure and collect real-time data of equipment through sensors. Classification of sensor data with the aid of AI or statistical techniques is the most basic building block of PdM activities in order to estimate time of the failure or RUL of the equipment [17–20].

Businesses frequently use the AI techniques and Internet of Things (IoT) to implement PdM activities in their equipment or operations.

The use of AI in PdM activities can adapt routine maintenance activities to the needs of each piece of equipment in the system. AI can be trained using past failures and their data, and predict the timing of future failures. AI can automatically detect anomalies in equipment and provide a quick prediction of when equipment will fail, preventing unexpected interruptions in production.

PdM activities have been used frequently and are a popular concept. PdM forecasts have many benefits such as minimizing unexpected outages, increasing equipment efficiency, minimizing costs by avoiding unnecessary maintenance, maximizing actual production time, reducing the number of breakdowns and increasing occupational safety.

PdM is a major part of the industry that requires periodic engine maintenance, in the same way as the aeronautics, automotive and railway industry. It is crucial to prepare the engines' maintenance schedule and develop a management strategy in order to maximize efficiency and safety. In PdM, generally the sensor data from the engines must be used to estimate the RUL [21–24].

In this study, literature studies on AI techniques and PdM activities in the transportation systems and spare parts sector were examined. The structure of this study is as follows: the main logic, constraints, and strategy of SLR are described in Section 2. Section 3 describes how SLR is carried out and how studies are classified according to different factors. Lastly, in Section 4, the conclusions achieved by this study are highlighted, and remarkable results are presented.

## 2. Main Framework of the Systematic Literature Review

Systematic literature review (SLR) is a strategy used to evaluate the important parts of the literature for a specific field. SLR may assist the study's aims, by pointing out the studies of interest with similar scopes, appraising them fundamentally in their techniques and putting them together in a measurable format when they can make a contribution [25,26].

Although there are other methods than SLR for summarizing the literature, such as bibliometric methods [27] or discrete choice methods [28–33], SLR has been preferred because it has become popular in recent years and does not require any extra software for its application. All of the mentioned methods have the potential to introduce a systematic, transparent and reproducible review process and thus improve the quality of reviews.

### 2.1. Systematic Literature Review Application Method

This study uses the following method for the SLR:

- Research questions:

  RQ1. What is the trend of PdM in the transportation sector in the last 5 years?
  RQ2. What are the fields of the publishers' that have published PdM studies?
  RQ3. Where are the PdM studies usually indexed?
  RQ4. Which transportation fields are the PdM techniques widely used?
  RQ5. Which data are used to apply PdM techniques? (Inputs).
  RQ6. Which algorithms/methods are used to apply PdM techniques?
  RQ7. Which results were expected from PdM applications? (Outputs).

- Literature survey databases: Well-known scientific databases used for literature survey, which are IEEE Xplore, ResearchGate, ScienceDirect and YokTez (for theses).
- Inclusion criteria:

  I1. Studies on the subject of PdM in transportation systems.
  I2. Studies published between 2017 and 2022 (filtering the 5 year period).
  I3. Studies which are research articles, conference papers and theses.
  I4. Studies that have an English version.
  I5. Studies which are four or more pages long.

- Exclusion criteria:

  E1. Studies unrelated to PdM in transportation systems.
  E2. Studies made before 2017.
  E3. Studies which are books, technical reports, reviews and commentary.
  E4. Studies that do not have an English version.
  E5. Studies which are less than four pages long.

### 2.2. Database Survey Strings

The survey was conducted from 9 June to 16 June 2022. For the SLR application, specific search stings formulated and applied on each database (IEEE Xplore, ResearchGate, ScienceDirect, YokTez) as follows:

- String 1: "Predictive Maintenance" and "Transportation" or "Transport"
- String 2: "Predictive Maintenance" and "Automotive" or "Automobile"
- String 3: "Predictive Maintenance" and "Aircraft" or "Aeronautic" or "Jet Engine"
- String 4: "Predictive Maintenance" and "Railway" or "Train" or "Wagon"
- String 5: "Predictive Maintenance" and "Marine" or "Maritime" or "Ship"
- String 6: "Predictive Maintenance" and "Vehicle"

## 3. Systematic Literature Review

SLR is a method that systematically examines, classifies and summarizes previous studies in the literature for a specific subject [34–36]. A SLR should be supported by figures and tables and made visually understandable. In this section, selected studies are analyzed

and classified from different perspectives [37–40]. The classifications made are shown with graphics and supported by numerical results.

Table 1 summarizes the studies reviewed in this SLR before proceeding to the SLR section. It provides a summary of the primary information about the transportation fields, equipment/case, methods/algorithms, goals, publication types and the general framework which is the starting point for the SLR. It gives a preliminary idea of what information will be used in the SLR. The abbreviations used in this study are given in Abbreviations.

The quantity of searched papers in the databases by using the preferred survey strings is shown in Figure 2.

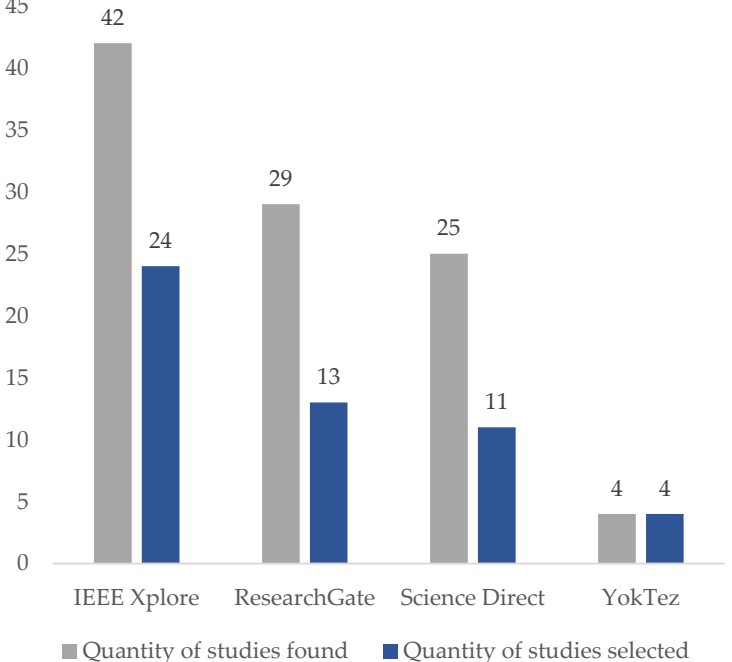

**Figure 2.** Quantity of searched papers in the databases.

The total quantity of found studies was 96. The quantity of the studies selected for this SLR is 52. A total of 44 of the found studies were rejected by using the exclusion criteria E1–5.

The IEEE Xplore, ResearchGate and Science Direct databases are some of the most well-known databases in the academic community. In addition, YokTez is a database containing MSc and PhD theses in both the English and Turkish language. Since there may be PdM applications in theses, we did not exclude theses in this SLR.

In Figure 3, there is a pie chart which shows the distribution by three main publication types. Among the selected studies, 52% were published by the journal type, while 40% were published by the conference paper type. It can be seen that 8%, which is very small, consists of theses.

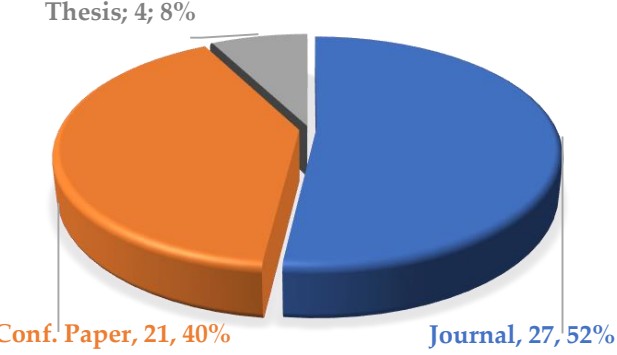

**Figure 3.** Distribution by publication types.

**Table 1.** Summarization of the studies reviewed in SLR.

| Ref. | Transp. Field | Equipment/Case | Method/Algorithm | Goal/Output | Publ. * |
|---|---|---|---|---|---|
| [41] | Aeronautics | Aircraft engine | ML, DL, LSTM | RUL | CP |
| [42] | Aeronautics | Aircraft equipment | Dig. Twin | DT integration | CP |
| [43] | Aeronautics | Aircraft equipment | MLP, SVR, LR, GA, DE | Fault classification | PhD |
| [44] | Aeronautics | Aircraft equipment | SVM, k-Means, k-NN, ARIMA, RVM | RUL | CP |
| [45] | Aeronautics | NASA's C-MAPSS | ANN | Fault diagnosis | CP |
| [46] | Aeronautics | NASA's C-MAPSS | RF, DL | Fault diagnosis | J |
| [47] | Aeronautics | NASA's C-MAPSS | GRU, LSTM, RNN, DL | RUL | CP |
| [48] | Aeronautics | NASA's C-MAPSS | ML, DL, LSTM, I4.0 | RUL | CP |
| [49] | Aeronautics | NASA's C-MAPSS | ML, LR, RF | RUL | CP |
| [50] | Aeronautics | NASA's C-MAPSS | RF, GB | RUL | CP |
| [51] | Aeronautics | NASA's C-MAPSS | LSTM | RUL | J |
| [52] | Aeronautics | NASA's C-MAPSS | LSTM, Mathematical Programming | RUL | J |
| [53] | Aeronautics | NASA's C-MAPSS | LSTM, SVM | RUL | J |
| [54] | Aeronautics | NASA's C-MAPSS | ML, DL, LSTM, ANN | RUL | J |
| [55] | Aeronautics | NASA's C-MAPSS | LSTM, LR, k-Means, SVM | RUL | MSc |
| [56] | Automotive | Automobile crane | ML, IoT, I4.0 | Fault diagnosis | CP |
| [57] | Automotive | Automobile maint. | ML ANN | Fault diagnosis | J |
| [58] | Automotive | Automobile maint. | Dig. Twin, Simulation | RUL | CP |
| [59] | Automotive | Automobile maint. | LSTM, DL | RUL | J |
| [60] | Automotive | Automobile maint. | ML, Time Series | RUL | J |
| [61] | Automotive | Engine | SVM, DT, ANN | Fault diagnosis | CP |
| [62] | Automotive | Engine | RF, NN, SVM, GP | Fault diagnosis | J |
| [63] | Automotive | Fleet management | DCNN, NB, k-Means | Fault diagnosis | J |
| [64] | Automotive | Fleet management | LSTM, DCNN, RNN, ANN, SVM | Fault diagnosis | J |
| [65] | Automotive | Gearbox | Dig. Twin | RUL | J |
| [66] | Maritime | Cruiser maintenance | ML, DL, LR | Fault diagnosis | CP |
| [67] | Railway | Air compressor | DL, LR, Time Series | Fault diagnosis | CP |
| [68] | Railway | Switch machine | Dig. Twin, LSTM, ARIMA, IoT, I4.0 | DT integration | J |
| [69] | Railway | Train maintenance | ML | RUL | CP |
| [70] | Railway | Train maintenance | Agents | RUL | J |
| [71] | Railway | Wheels | ANN | RUL | J |
| [72] | Vehicle parts | Battery | EMD, GRA, RNN, LSTM | RUL | J |
| [73] | Vehicle parts | CNC machines | Dig. Twin, Simulation | DT integration | J |
| [74] | Vehicle parts | CNC machines | LSTM, CNN, ARIMA, RNN | Fault diagnosis | J |
| [75] | Vehicle parts | CNC machines | LSTM, DL | Fault diagnosis | J |
| [76] | Vehicle parts | CNC machines | LSTM, DL | RUL | J |
| [77] | Vehicle parts | Electrical equipment | Dig. Twin | DT integration | CP |
| [78] | Vehicle parts | Electrical equipment | ML, ANN | Fault diagnosis | J |
| [79] | Vehicle parts | Engine | ML, ANN | Fault classification | CP |
| [80] | Vehicle parts | Engine | Agents, DL | Fault diagnosis | CP |
| [81] | Vehicle parts | Engine | LSTM, DL, ANN | Fault diagnosis | J |
| [82] | Vehicle parts | Industrial robot | Dig. Twin | RUL | CP |
| [83] | Vehicle parts | Industrial robot | Dig. Twin, Simulation | RUL | J |
| [84] | Vehicle parts | Pump | ML, Deep Leaning, ANN, LSTM | Fault diagnosis | MSc |
| [85] | Vehicle parts | Pump | DL, EMD, NN | RUL | J |
| [86] | Vehicle parts | Roller | SVM, RF, DT, k-Means | Fault classification | J |
| [87] | Vehicle parts | Roller | Softmax, k-Means, SVM, DT, NB | Fault diagnosis | PhD |
| [88] | Vehicle parts | Roller | ML, DL | RUL | J |
| [89] | Vehicle parts | Semiconductor | k-NN, LR, RF | RUL | CP |
| [90] | Vehicle parts | Semiconductor | ML, DL, ANN, RF, GB | RUL | CP |
| [91] | Vehicle parts | Vehicle parts | Dig. Twin | Fault diagnosis | CP |
| [92] | Vehicle parts | Vehicle parts | RF, GB, AdaBoost, MLP, SVR | RUL | J |

* J: Journal; CP: Conference Paper; MSc: MSc Thesis; PhD: PhD Thesis.

The percentage of studies in journal type and the percentage of studies in conference paper type were close to each other. From this point of view, it can be deduced that the PdM in transportation topic has been trending in academic events in the last 5 years and that studies are ongoing.

### 3.1. Answers to RQ1: The Trend of the PdM in Transportation Sector in Last 5 Years

In Figure 4, a histogram is shown of the annual growth of publications by years. In addition, there is a highlighted increasing trend curve which showing an interest in PdM studies in transportation over the recent years. With the technologic evolution in the motor vehicles industry, the interest in PdM techniques has increased even more. All transportation vehicles consist of complex parts and components. The lifetime of each element is different. For this reason, when examining a transportation vehicle, it is necessary to consider each element separately, not as a whole. This increases the need for PdM techniques day by day.

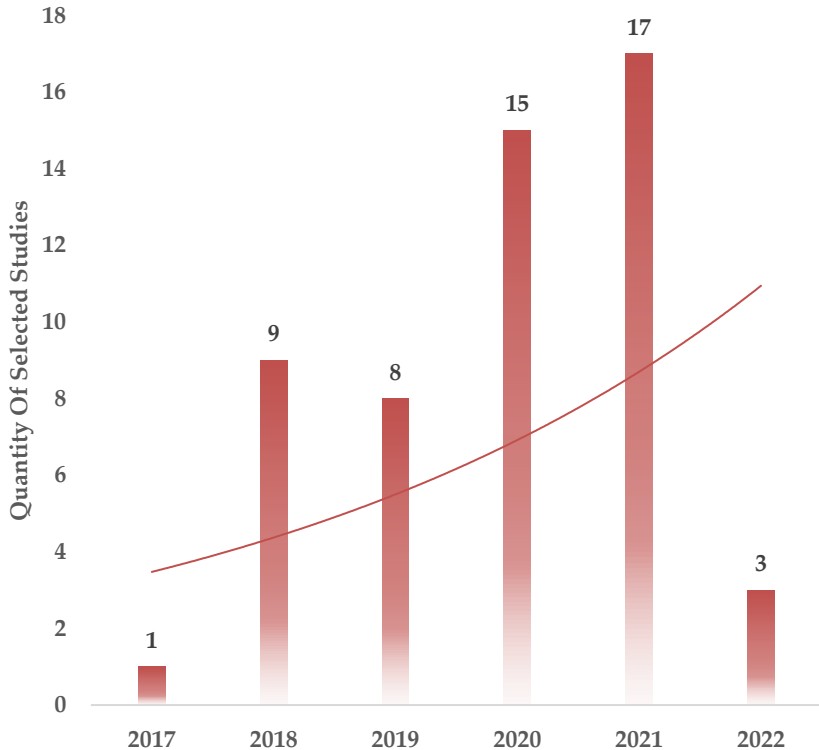

**Figure 4.** Annual growth of publications by years.

Obtaining an answer to RQ1 is important for researchers who will conduct PdM studies in the transportation sector to see the direction of the trend. The fact that there has been an increasing trend in the last 5 years shows that the studies to be completed in this field are gaining importance and that it is an area open to development.

### 3.2. Answers to RQ2: Distribution of Studies by Publishers' Fields

In Table 2, the studies that are shown belong to 27 journals. Among them, the *Advanced Engineering Informatics*, *Computers & Industrial Engineering*, *Computers in Industry*, *Procedia Manufacturing*, *Reliability Engineering & System Safety* and *Sensors* journals have two publications. Other journals have only one publication.

The majority of the selected journals operate in the field of engineering and computer science. Therefore, it can be deduced that PdM techniques are in vogue in engineering.

In Figure 5, there is a word cloud concerning the publishing journals. A word cloud, also known as a tag cloud, is a visualization technique that shows how frequently words appear in a disordered text, by adjusting the size of each word according to its frequency. All the words are then ordered in a cluster of words. The more often the word repeats, the larger its size.

**Table 2.** Distribution of publication year(s) by publishing journals.

| Journal Title | Publication Year(s) | |
|---|---|---|
| *Advanced Engineering Informatics* | 2020 | 2021 |
| *Computers & Industrial Engineering* | 2021 | 2021 |
| *Computers in Industry* | 2021 | 2022 |
| *Procedia Manufacturing* | 2020 | 2020 |
| *Reliability Engineering & System Safety* | 2021 | 2022 |
| *Sensors MDPI* | 2021 | 2021 |
| *Electronics MDPI* | 2021 | |
| *Energies MDPI* | 2017 | |
| *Expert Systems with Applications* | 2021 | |
| *Forschung im Ingenieurwesen* | 2021 | |
| *IEEE Access* | 2021 | |
| *IEEE/CAA Journal of Automatica Sinica* | 2021 | |
| *Information MDPI* | 2021 | |
| *International Journal of Advanced Manufacturing Technology* | 2021 | |
| *International Journal of Computer Integrated Manufacturing* | 2019 | |
| *Journal of Information Technologies (JIT)* | 2019 | |
| *Journal of Intelligent Manufacturing* | 2020 | |
| *Materials Today: Proceedings* | 2022 | |
| *Procedia CIRP* | 2019 | |
| *Proceedings MDPI* | 2020 | |
| *Robotics and Computer-Integrated Manufacturing* | 2020 | |

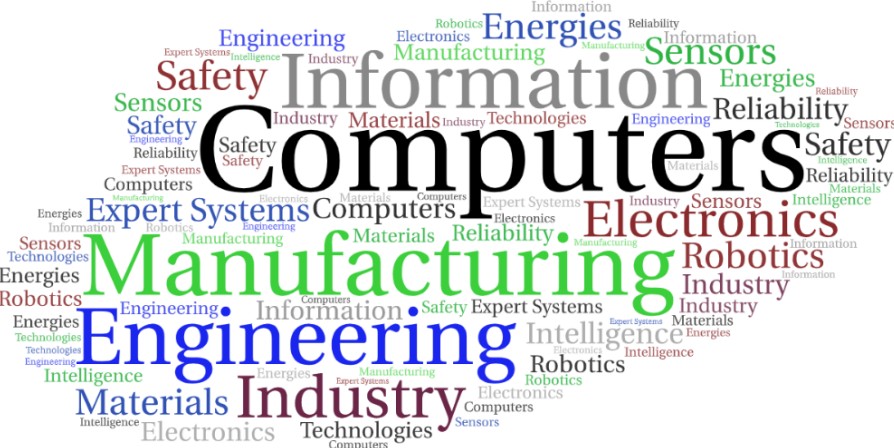

**Figure 5.** Publishing journals' word cloud.

It can be seen in Figure 5 that some of the words are larger than others. Based on this, it can be deduced in which journals' topic of operation the PdM techniques are of more interest. The words "Computers", "Manufacturing", "Engineering", "Information" and "Electronics" came to the fore. It can be said that PdM techniques are used more widely in the journals operating in these fields.

In Table 3, the studies shown belong to 21 conference papers. Among them, the Int. Conf. on Data Science and Advanced Analytics and Int. Conf. on Emerging Technologies and Factory Automation (ETFA) conferences have two publications. The other conferences have only one publication. The majority of the selected conferences operate in the field of computer science, transportation, communication and data science.

It can be seen in Figure 6 that some of the words are repeated more than others. Based on this, it can be deduced in which conference field the PdM techniques are of more interest. The words "Transportation", "Computing" and "Electronics" came to the fore. It can be said that PdM techniques are used more widely in conferences held in these fields.

**Table 3.** Distribution of publication year(s) by publishing conferences.

| Publishing Conference | Publication Year(s) | |
|---|---|---|
| Int. Conf. on Data Science and Advanced Analytics | 2018 | 2021 |
| Int. Conf. on Emerging Technologies and Factory Automation (ETFA) | 2018 | 2020 |
| ACM/SIGAPP Symposium on Applied Computing | 2019 | |
| AIP Conference Proceedings | 2018 | |
| CIRP Conference on Manufacturing Systems | 2019 | |
| Innovations in Intelligent Systems and Applications Conference (ASYU) | 2019 | |
| Int. Conf. on Big Data (Big Data) | 2018 | |
| Int. Conf. on Electrical, Electronics, Comm., Comp. Tech. and Opti. Technq. | 2018 | |
| Int. Conf. on ICT for Smart Society (ICISS) | 2021 | |
| Int. Conf. on Information and Communication Technology Convergence | 2020 | |
| Int. Conf. on Intelligent Transportation Systems (ITSC) | 2020 | |
| Int. Conf. on Mathematics and Mathematics Education (ICMME 2021) | 2020 | |
| Int. Conf. on Recent Trends In Advanced Computing 2019 | 2019 | |
| Int. Conf. on Smart Computing (SMARTCOMP) | 2019 | |
| Int. Conf. on Telecommunications and Signal Processing | 2021 | |
| International Symposium on NDT in Aerospace | 2018 | |
| IOP Conference Series: Materials Science and Engineering | 2020 | |
| Workshop on Microelectronics and Electron Devices (WMED) | 2018 | |
| World Forum on Internet of Things (WF-IoT) | 2020 | |

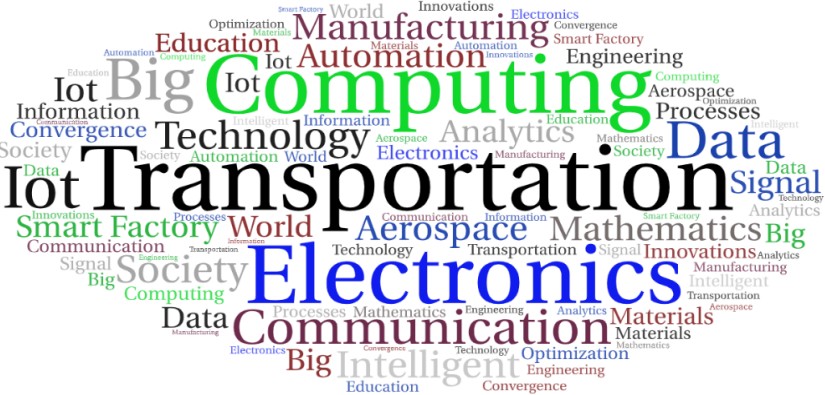

**Figure 6.** Publishing conferences' word cloud.

Obtaining an answer to RQ2 is important for researchers who will conduct PdM studies to see the publishers' working fields. The fact that there are transportation systems, computer systems, manufacturing systems, engineering, information systems and electronics in the work fields shows that the studies to be completed in these fields are gaining importance and these fields are open for development.

*3.3. Answers to RQ3: Distribution of Studies by Journals' Indexing*

A total of 21 of the selected studies are conference papers and these are not included in this subsection. In addition, 27 of the selected studies were published in the journals and these journals' indexes are examined in this subsection.

The major indexes of the 27 journals are determined as follows: 17 of the journals belong to the Science Citation Index Expanded (SCIE), 5 of the journals belong to the Science Citation Index (SCI), 3 of the journals belong to the Scopus Index, 1 journal belongs to the Inspec Index and 1 journal belongs to the Ulakbim Index.

While determining the major indexes, the most respected and known scientific indexes in the academic community were used. The Science Citation Index (SCI), Science Citation Index Expanded (SCIE) and Emerging Sources Citation Index (ESCI) are the most well-known of these. If a journal is indexed in one of these indexes, its major index is assigned

as SCI, SCIE or ESCI. If a journal is not indexed in one of these indexes, other options were considered in order to determine the major index. As the order of viewing, a sequence such as Scopus, Directory of Open Access Journals (DOAJ), Inspec, Ebsco, Ulakbim, Proquest, etc., was applied, respectively.

Answering RQ3 is important to provide a preview for researchers interested in studying PdM. A journal's indexing is mostly important for the academics. For this reason, it is an important advantage for the academics who will conduct a PdM study in the transportation sector to predict their studies' potential indexing in the future.

### 3.4. Answers to RQ4: Distribution of Studies by Different Transportation Fields

The quantity of selected studies according to different fields within the transportation sector is shown in Figure 7. Vehicle parts are seen as the most frequent field of studies, followed by aeronautics and automotive fields.

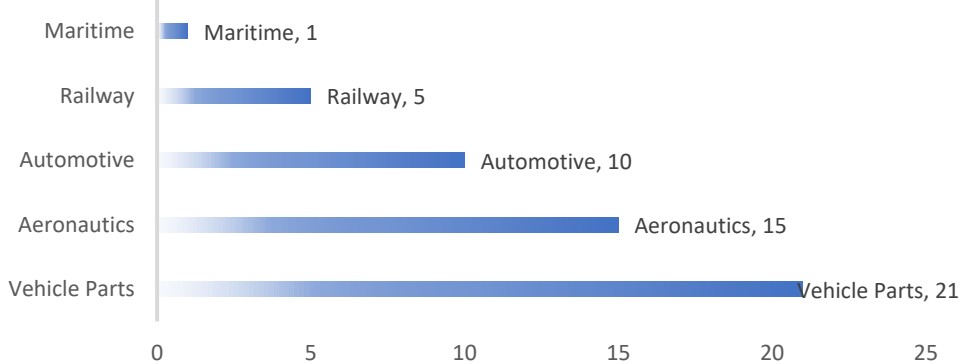

**Figure 7.** Quantities by different transportation fields.

When the literature on PdM techniques in the motor vehicle industry is examined, many studies may seem unrelated to the transportation sector at first glance. The reason for this is that vehicles used in transportation are not always considered as a whole. Transportation vehicles are very complex and consist of many subparts. Examples of these subparts are engine parts, gears, bearings, coils, spark plugs and gearboxes which are universal parts that can be used in all types of vehicles with different shapes and calibers.

Each of these subparts is crucial for the operating of the vehicle, and a malfunction that may occur in one of these parts may result in inoperability of the vehicle. Even worse, this subpart malfunctions can go as far as stopping the vehicle while in motion, causing an accident and even causing death. For this reason, most of the studies in the motor vehicle industry have actually focused on the PdM of each of the subpart separately. Therefore, we have included the vehicle parts' field separately in our study.

Obtaining an answer to RQ4 is important for researchers who will conduct PdM studies in transportation systems to see which sectors are gaining focus in the transportation aspect. The fact that studies are conducted mostly in the aeronautics and automotive sectors shows that the studies to be completed in these sectors are gaining importance in recent years and these sectors need more studies to be completed. It is also noteworthy that the number of studies in the railway sector is low. It can be concluded that the literature needs more studies carried out in the railway sector.

### 3.5. Answers to RQ5: Distribution of Studies by Input Parameters and Sensors

It can be said that PdM is applied to the most varied equipment in the most varied fields. This might be due to the specific attributes of each PdM case. In general, synthetic data cannot represent a real event or failure, and generating synthetic data requires knowledge of the equipment [93–97].

As seen in Figure 8, no synthetic data were used in any of the selected studies. The datasets used in all studies are the real data of the equipment or system. Fault classification

or DT integration were performed in 8 of the 52 studies examined within the scope of SLR. For this reason, the number of studies using a tangible input has been determined as 44. While sensors were used in 27 of the studies to obtain the real data, the fault records kept in the past were used in 5 of them. In addition, one study using thermal camera images was found.

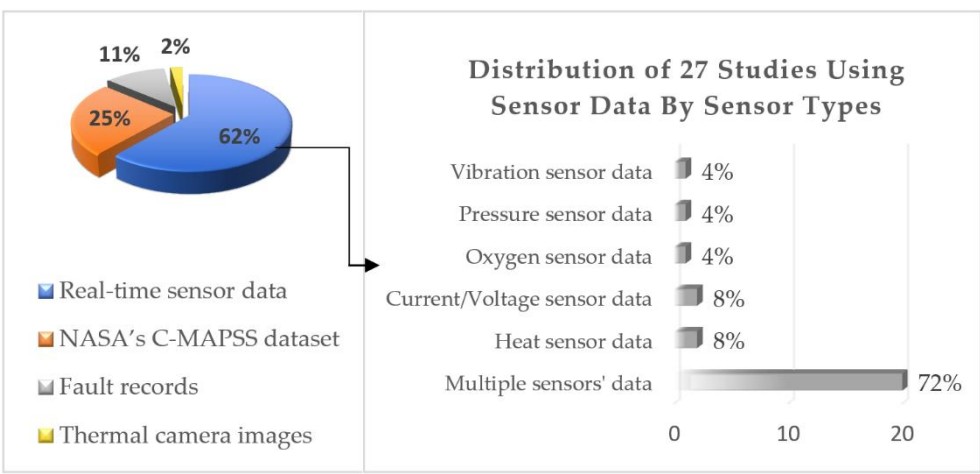

**Figure 8.** Quantities by different inputs.

The distribution of 27 studies using sensor data by sensor types are shown in the bar graph section. It has been observed that vibration, pressure, oxygen, current/voltage and heat sensors are used as sensor types. Studies using different types of sensors together are classified as multiple sensors' data and 20 of 27 studies were found to be included in this class.

Another remarkable point is that the NASA Turbofan Jet Engine Dataset also called NASA's C-MAPSS dataset is used very commonly in PdM studies. Even though that dataset was shared a long time ago, it remains popular and relevant in recent years. Several hundred new studies have been published from this dataset so far. These studies present and benchmark novel algorithms to predict the RUL of the mentioned jet engine. In this review, it was seen that 11 of the 44 studies that made a PdM application in the transportation sector used NASA's C-MAPSS dataset as input.

Obtaining an answer to RQ5 is important for researchers who will conduct PdM studies in transportation systems to see how the different input parameters can be used. The large number of studies using NASA's C-MAPSS dataset shows that there is competition relating to this dataset. Although there are many studies that a researcher who wants to work on this dataset can take as an example, there will also be many competitors. Researchers who do not want to participate in this competition can also conduct PdM studies using sensor data. As a matter of fact, the high number of studies using sensor data supports this argument.

### 3.6. Answers to RQ6: Distribution of Studies by Algorithms and Methods

When selected studies are evaluated, it can be seen that many different AI algorithms are used for PdM estimations. Some of the techniques used are regression-based methods, while the aim is to estimate the RUL, fault diagnosis, etc. In addition, some of the studies are designed for fault classification.

When PdM studies in the transportation sector and spare parts are examined, it was determined that AI techniques, heuristics and mathematical models were used. It has been determined that 29 different algorithm types are used in total and the frequencies of the nine most frequently used algorithms and methods are shown in Figure 9.

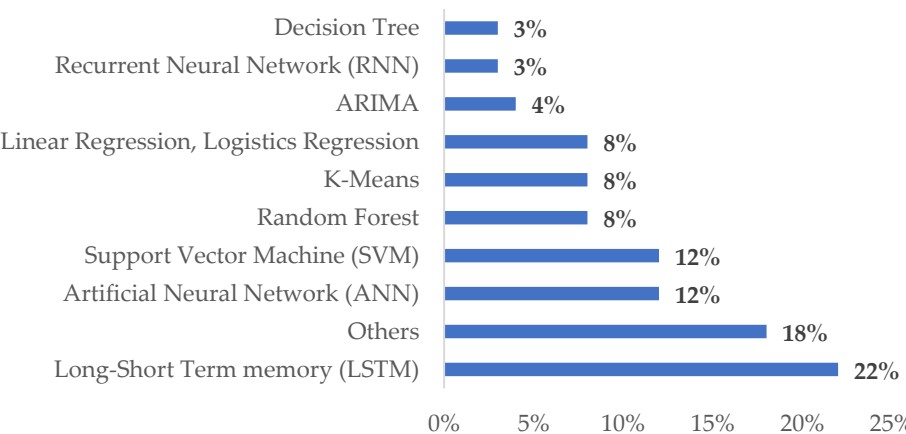

**Figure 9.** Frequently used algorithms and methods.

It is seen that the most commonly used algorithm in PdM estimations is the LSTM networks. The main reason for this situation is related to the nature of the problem. LSTM networks are networks that work with high efficiency and are often preferred in cases such as determination of long-term dependencies and time series analysis. For this reason, the LSTM structure was preferred in many studies selected.

In the studies, the ANN was also used for time series analysis in general, similar to the LSTM structure. The ANN, which has a usage rate of 12%, is generally preferred for comparison with other model results.

SVM, which has a rate of 12%, is among the machine learning techniques which used for regression analysis and classification. The SVM technique is followed by Random Forest (RF) and K-Means algorithms, which are different classification models.

In the selected studies, it has been observed that it is used in heuristics and mathematical models as well as AI models such as Softmax, CNN, Gradient Boosting, AdaBoost, RVM, GRU under the category of "Others", which has a 18% percentage.

Answering RQ6 is important to summarize the previously used algorithms or methods for researchers interested in studying PdM. The high number of studies using LSTM, ANN or SVM shows that the subject of PdM in the transportation sector is saturated with these methods. It shows that researchers who want to study PdM in the transportation sector can reduce the originality of the study if they use these methods, or if it is necessary to use these methods, they should definitely add an improvement suggestion to these methods.

### 3.7. Answers to RQ7: Distribution of Studies by Output Parameters

Since PdM processes are stimulating processes and are carried out with different estimation methods, they are a process that must be optimized in order to minimize the maintenance cost and achieve zero defect service, depending on the accuracy of the estimation.

PdM processes are followed by three basic steps. First of all, the vibration and frequency data of the machine are collected at certain periodic times in order to follow the situation and the inputs are determined. Then, different models and algorithms are run in order to process the data and determine the outputs by determining the performance information of the machine. In the last stage, maintenance processes are planned and put into use in line with the outputs obtained.

Within the scope of the selected studies, the inputs related to the motor vehicle industry and spare parts sectors and the models used are explained in the previous sections.

Although there are many different inputs in the studies, the outputs are generally concentrated in four different categories. These categories are fault diagnosis, fault classification, RUL and digital twin integration. The frequencies of the outputs are shown in Figure 10.

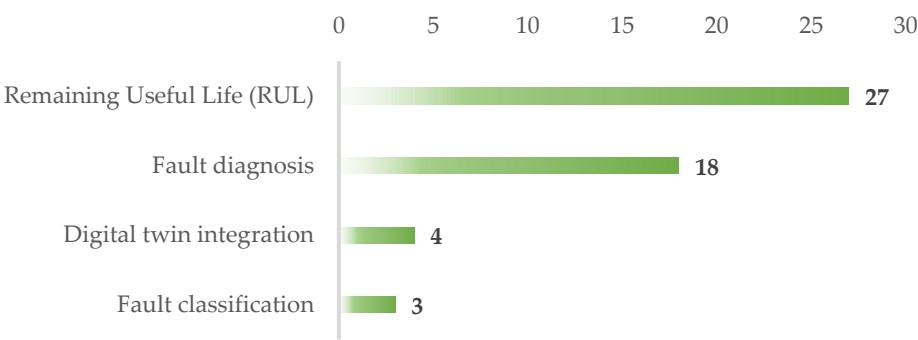

**Figure 10.** Frequencies by different outputs.

Most studies have made RUL estimations. While making RUL estimations, LSTM, ANN and regression models were generally used.

Studies on fault diagnosis and fault classification have similar characteristics and are basically analyzed within the scope of SVM, RF, k-Means and different DT models.

Outputs categorized as digital twin integration also emerge with the concept of I4.0. These studies are carried out in order to create inputs for different studies together with the concepts of the IoT and smart production systems. In these studies, advantages of the digital twin concept and the technological improvements that can be used in terms of obtaining large and meaningful data about the process are mentioned.

Obtaining an answer to RQ7 is important for researchers who will conduct PdM studies in transportation systems to see the different output parameters or goals that can be used. If the study to be carried out is an application study, there are two basic goals in the literature: RUL estimation or fault diagnosis. Researchers should choose one of these two main objectives for their application. Along with I4.0, DT integration studies are also seen in the literature in recent years. The low number of studies in this area is an opportunity for researchers.

## 4. Conclusions and Discussion

The use of modern transportation systems requires the adoption of a good engineering approach and the implementation of appropriate and timely maintenance strategies in order to keep the system in top working condition. PdM, which is one of the maintenance strategies, focuses on collecting and evaluating data from the sensors and reaching an estimated result about when maintenance will be performed. The aim is to ensure that the equipment operates at high performance by intervening before a malfunction occurs.

In this SLR we conducted a systematic review and analysis of 52 studies to answer the following RQs:

RQ1. What is the trend of PdM in the transportation sector in the last 5 years?

RQ2. What are the fields of the publishers' that have published PdM studies?

RQ3. Where are the PdM studies usually indexed?

RQ4. Which transportation fields are the PdM techniques widely used?

RQ5. Which data are used to apply PdM techniques? (Inputs).

RQ6. Which algorithms/methods are used to apply PdM techniques?

RQ7. Which results were expected from PdM applications? (Outputs).

The main conclusions of the SLR are summarized below:

As the conclusion to RQ1, it can be said that there was an increasing trend in PdM studies in the transportation sector between 2017 and 2022 (see Section 3.1). Due to the continuous development of technology and the reflection of these developments on the transportation sector, there has been an exponential increase in PdM studies since 2017. According to that, the importance of using the PdM technique in maintenance activities is increasing day by day and this technique is being used more and more widely in the transportation sector. Popularity of the PdM technique is growing in the transportation sector.

As the conclusion to RQ2, publishers who have published PdM studies in the last 5 years were mostly related to transportation systems, computer systems, manufacturing systems, engineering, information systems and electronics (see Section 3.2). It can be said that publishers operating in these fields were more interested in PdM studies. If a PdM study is desired to be carried out in the future, it is preferred by the publishers that the study is related to one of the fields mentioned above.

As the conclusion to RQ3, 27 out of 52 studies examined within the scope of SLR were indexed in international indexes (see Section 3.3). Other studies were not included in an index as they are conference papers and theses. A total of 22 of the 27 studies were indexed in SCI or SCIE indexes, which corresponded to 81.5% proportionally.

RQ3 was included in this study to provide a preview for academics interested in studying PdM. For academics, a journal's indexing is necessarily important. For this reason, it is an important advantage for the academics who will conduct a PdM study in the transportation sector to predict their studies' potential indexing in the future.

As the conclusion to RQ4, the trend towards automotive and aeronautics in terms of motor vehicle industry has increased in recent years (see Section 3.4). When PdM studies carried out in the transportation sector between 2017 and 2022 were examined, it was seen that 88.4% were related to aeronautics, automotive and their spare parts together. It has been observed that computational experiments on PdM studies in the transportation sector are mostly carried out on equipment used in transportation vehicles such as motors, gears, bearings and coils. PdM applications focus on monitoring the status of these equipment, evaluating their performance, RUL estimation, fault diagnosis and detection. It was also seen that PdM studies in the maritime sector were insufficient and it is an area open to research.

As the conclusion to RQ5, within the 52 studies, 44 were found to use a tangible input (see Section 3.5). Real-time sensor data were used in 27 of these 44 studies, which corresponds to a proportional ratio of 61.4%. In addition, NASA's C-MAPSS dataset was used in 11 of these 44 studies, which corresponded to 25% proportionally. It is seen that sensor data and NASA's C-MAPSS dataset are very frequently used in PdM studies.

The number of studies using data from more than one sensor type at the same time was determined as 20. In terms of proportion, 20 of the 27 studies using sensor data used multiple sensors' data, which corresponds to 72%. It can be said that in most of the studies using sensor data, a single sensor type was not adhered to and different sensors were used together.

As the conclusion to RQ6, it was seen that LSTM, ANN, SVM, RF, k-Means, LR, ARIMA, RNN and DT were used most frequently for PdM applications (see Section 3.6). It is seen that the most commonly used algorithm in PdM estimations is the LSTM networks with the percentage of 22%. LSTM networks are networks that work with high efficiency and are often preferred in cases such as determination of long-term dependencies and time series analysis. This may be the reason why the LSTM method was preferred in most of the studies. On the other hand, ANN was also used for time series analysis in general, similar to the LSTM structure. The ANN, which has a usage rate of 12%, was commonly used for comparison with other models. Another frequently used model was SVM, which has a rate of 12%, is among the machine learning techniques which used for regression analysis and classification. In addition, it has been observed that it was used in heuristics and mathematical models as well as AI models such as Softmax, CNN, Gradient Boosting, AdaBoost, RVM, GRU under the category of "Others", which has a 18% percentage.

As the conclusion to RQ7, the outputs were concentrated in four different categories. These categories were RUL estimation, fault diagnosis, fault classification and DT integration (see Section 3.7). A total of 27 out of 52 studies made RUL estimations which had a rate of 51.9%. While making RUL estimations, LSTM, ANN and regression models were generally used. A total of 18 out of 52 studies made fault diagnoses and 3 out of 52 studies made fault classifications, which had a total rate of 40.3%. Studies on fault diagnosis and fault classification had similar characteristics and were generally analyzed within the scope

of SVM, RF and k-Means. A total of 4 out of 52 studies made DT integration. Outputs categorized as DT integration also merged with the concept of I4.0. These studies were carried out in order to create inputs for different studies together with the concepts of the IoT and smart production systems. In these studies, the advantages of the DT concept and the technological improvements that can be used in terms of obtaining large and meaningful data about the processes were covered.

To summarize the conclusions, there has been a large increase in PdM studies, which are related to the transportation sector, between 2017 and 2022. According to the studies examined, it is seen that AI techniques have been used intensively in PdM estimations in spare parts, machinery and equipment depending on the transportation sector for the last 2 years. In addition, it is seen that the concepts of IoT and smart production systems, for which the concept of I4.0 has increased its popularity, are frequently used in many PdM studies.

In the examined studies, it was observed that the AI techniques that are widely used for PdM estimations are often performed with experimental sets or simulation data. Studies show that AI techniques produce meaningful results for PdM estimations. Therefore, it is thought that PdM techniques will not only remain in academic studies, but will also be used practically in the real world, especially in the motor vehicle industry.

Researchers who will work on PdM in the future will be able to contribute to the literature by including concepts such as DT, cloud technology, Big Data and IoT in the maintenance models they will design in the next stage.

**Author Contributions:** Conceptualization, O.Ö.E. and A.F.İ.; methodology, A.A.; software, A.F.İ.; validation, A.A., A.K.T. and S.E.; formal analysis, O.Ö.E.; investigation, A.F.İ.; resources, S.E.; data curation, O.Ö.E.; writing—original draft preparation, A.F.İ.; writing—review and editing, A.F.İ.; visualization, O.Ö.E.; supervision, A.A. and S.E.; project administration, S.E. All authors have read and agreed to the published version of the manuscript.

**Funding:** This research received no external funding.

**Institutional Review Board Statement:** Not applicable.

**Informed Consent Statement:** Not applicable.

**Data Availability Statement:** Not applicable.

**Conflicts of Interest:** The authors declare no conflict of interest.

## Abbreviations

List of Abbreviations

| | |
|---|---|
| AI | Artificial Intelligence |
| ARIMA | Autoregressive Integrated Moving Average |
| CNN | Convolutional Neural Network |
| DCNN | Deep Convolutional Neural Network |
| DE | Differential Evolution |
| DL | Deep Learning |
| DT | Decision Tree |
| EMD | Empirical Mode Decomposition |
| GA | Genetic Algorithm |
| GB | Gradient Boosting |
| GP | Gaussian Processes |
| GRA | Grey Relationship Analysis |
| GRU | Gated Recurrent Unit |
| I4.0 | Industry 4.0 |
| IoT | Internet of Things |
| k-NN | k-Nearest Neighbors |
| LR | Linear Regression |
| LSTM | Long Short-Term Memory Network |

| ML | Machine Learning |
| MLP | Multi-layer Perceptron |
| NB | Naïve Bayes |
| PdM | Predictive Maintenance |
| RF | Random Forests |
| RNN | Recurrent Neural Network |
| RUL | Remaining Useful Life |
| RVM | Relevance Vector Machine |
| SLR | Systematic Literature Review |
| SVM | Support Vector Machine |
| SVR | Support Vector Regression |

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
