# Peer review of "A Systematic Literature Review of the Predictive Maintenance from Transportation Systems Aspect"

_sustainability, doi:10.3390/su142114536_

Round 1
Reviewer 1 Report
A) General Considerations
The work entitled A Systematic Literature Review of the Predictive Maintenance from Transportation Systems Aspect provides a systematic literature review (SLR) of the predictive maintenance (PdM) techniques in transportation systems. The authors focus on PdM in the motor vehicles industry in the last 5 years. The work summarizes statistical, stochastic, and AI approaches for PdM applications and their goals, methods, findings, challenges, and opportunities. Finally, the authors encourage future research by indicating the areas that have not yet been studied on PdM. The work is very interesting. The document is well-written and in clear language. The structure of the document is adequate, and the conclusions of the work are aligned with the objectives initially outlined.
In the Introduction (1) section the authors indicate their motivation for the work, the object and objectives of the work, and indicate some of the methodological steps they intend to follow to achieve the objectives. Also, they introduce briefly concepts such as SLR, PdM, (and different types of maintenance), and AI.
In section (2), Main Framework of the Systematic Literature Review, the authors underline the concept and techniques of the Systematic literature review (SLR): Application Method, and Database Survey Strings.
In section (3) about Systematic Literature Review, the authors analyze and classify selected studies from different perspectives, giving a preliminary idea of what information will be used in the SLR. Then, they go deep into this field presenting in sequence several interesting distributions of studies: by Academic Databases; by Years; by Publication Types; by Publishing Journals and Conferences; by Journals’ Indexing; by Different Transportation Fields; by Input Parameters and Sensors; by Algorithms and Models; and by Output Parameters.
In the Conclusion and Discussion (4) section, the authors present and discuss several interesting conclusions; mainly: (A) AI techniques have been used intensively in PdM estimations in spare parts, machinery, and equipment depending on the transportation sector for the last 2 years; (B) AI techniques that are widely used for PdM estimations are often performed with experimental sets or simulation data; (C) Concepts of IoT and smart production systems, for which the concept of Industry 4.0 has increased its popularity, are frequently used in many PdM studies; and (D) Researchers who will work on PdM are encouraged to contribute to the literature by including concepts such as digital twin, cloud technology, big data and IoT in maintenance models they will design.
B) Details
Throughout the text, it is possible to find 84 bibliographic references, the last of which (84) is on line 261, page 10. However, in the final list of bibliographic references, there are 111. Where are they in the text?
Author Response
Dear reviewer,
Thank you very much for reviewing our study and sharing your valuable comments with us.
Although some studies are written in the bibliography, it has not escaped your notice that they are not cited in the text. This is because we are preparing another article on PdM. These studies have nothing to do with the transportation sector. These studies were added to the bibliography to form an outline and were inadvertently forgotten to be removed afterwards. We sincerely apologize for this unintentional mistake. You can be sure that we have made all the necessary corrections. Thank you for noticing this mistake and alerting us.
Also for your information, we have added new studies to our bibliography during the reviewing process and cited them in the text.
In addition, we made the following corrections in line with the suggestions from reviwers:
- We added the number of studies included in our SLR to the abstract.
- We added the inclusion criteria to the Section 2.1.
- We added the range of search period to the Section 2.2.
- We majorly revised the Section 3.
- At the end of each subsection in Section 3, we've added why the answer to that RQ is important, how it will benefit to the future research, and other noteworthy highlights. We tried to mention RQs at every stage of our study. We updated the order and names of the subsections in Section 3 to have the same layout as the RQs.
- We've majorly revised and improved significantly Section 4.
- We made sure the discussion part is in the same order as the RQs.
- We have summarized the conclusions reached along with the answer for each RQ.
- We discussed about the gaps in the literature, the points that future studies should focus on and areas with opportunity for improvement with the conclusion of each RQ.
- The remarkable points reached by the analyzes in Section 3 were mentioned in the discussion and the most important results were emphasized proportionally.
Thank you again for taking the time to share your valuable comments and suggestions with us. Please do not hesitate to write us if you have any further questions or suggestions.
Your's respectfully.
Reviewer 2 Report
Thank you for giving me an opportunity to review this paper. Overall, this is an interesting study. Nevertheless, there are some issues that must be improved before accepting for publication.
- To present a comprehensive review of study, the abstract should include a number of articles reviewed in this study.
- The study shows the exclusion criteria, but did not mention to the inclusion criteria.
- What was the range of search period of papers? (only survey date was presented)
- Papers mentioned about significance and processes of SLR should be more added to support the adoption of SLR method such as
- Critical Factors and Performance Measurement of Business Incubators: A Systematic Literature Review. Sustainability 2022, 14, 4610.
- A systematic literature review on software measurement programs. Information and Software Technology 2016, 73, 101-121.
- To better highlight the significant of maintenance management, some related papers should be included.
· DOI: 10.1016/j.cie.2020.106889
· DOI: 10.1109/IEEM.2011.6118030
- The most important weakness of the paper is the lack of motivation in each RQ. Why did the papers need to know the answers to each question? What are the advantages of obtained answers?
- The results should be presented as orders of RQ, and each part should mention to the number of RQ.
- The paper must focus more on the discussion part. Now, it mainly presented the results of literature review. It should be better to provide opinions on the obtained answers (suggestions, gaps, or improvement opportunities in each RQ). The discussion part is still weak and requires significant improvement.
Author Response
Dear reviewer,
Thank you very much for reviewing our study and sharing your valuable comments with us. We think that your suggestions are very correct and appropriate. Based on your suggestions, we made major corrections in our study. We provided a point-by-point response to your comments in MDPI format.
Point 1. To present a comprehensive review of study, the abstract should include a number of articles reviewed in this study.
Response 1. We added the number of studies included in our SLR to the abstract.
Point 2. The study shows the exclusion criteria, but did not mention to the inclusion criteria.
Response 2. We added the inclusion criteria to the Section 2.1.
Point 3. What was the range of search period of papers? (only survey date was presented)
Response 3. We added the range of search period to the Section 2.2.
Point 4. Papers mentioned about significance and processes of SLR should be more added to support the adoption of SLR method such as
-
- Critical Factors and Performance Measurement of Business Incubators: A Systematic Literature Review. Sustainability 2022, 14, 4610.
- A systematic literature review on software measurement programs. Information and Software Technology 2016, 73, 101-121
Response 4. We have examined the papers you have suggested and added them to our study, and we have broaden our analysis in line with following papers:
- Pattanasak P, Anantana T, Paphawasit B, Wudhikarn R. Critical Factors and Performance Measurement of Business Incubators: A Systematic Literature Review. Sustain. 2022;14(8). doi:10.3390/su14084610
- Tahir T, Rasool G, Gencel C. A systematic literature review on software measurement programs. Inf Softw Technol. 2016;73:101-121. doi:10.1016/j.infsof.2016.01.014
Point 5. To better highlight the significant of maintenance management, some related papers should be included.
- DOI: 10.1016/j.cie.2020.106889
- DOI: 10.1109/IEEM.2011.6118030
Response 5. We have examined the papers you have suggested. The first study (Zonta T, da Costa CA, da Rosa Righi R, de Lima MJ, da Trindade ES, Li GP. Predictive maintenance in the Industry 4.0: A systematic literature review. Computers and Industrial Engineering. 2020;150(August):106889. doi:10.1016/j.cie.2020.106889) was already available in our article before the review. We added the second study (Wudhikarn R. Implementation of Overall Equipment Effectiveness in Wire Mesh Manufacturing. IEEE International Conference on Industrial Engineering and Engineering Management, 2011, pp. 819-823. doi: 10.1109/IEEM.2011.6118030.) and we have highlighed the significance of maintenance.
Point 6. The most important weakness of the paper is the lack of motivation in each RQ. Why did the papers need to know the answers to each question? What are the advantages of obtained answers?
Response 6. At the end of each subsection in Section 3, we've added why the answer to that RQ is important, how it will benefit to the future research, and other noteworthy highlights. We tried to mention RQs at every stage of our study. We updated the order and names of the subsections in Section 3 to have the same layout as the RQs.
Point 7. The results should be presented as orders of RQ, and each part should mention to the number of RQ.
Point 8. The paper must focus more on the discussion part. Now, it mainly presented the results of literature review. It should be better to provide opinions on the obtained answers (suggestions, gaps, or improvement opportunities in each RQ). The discussion part is still weak and requires significant improvement.
Response 7 and 8. We've majorly revised and improved significantly Section 4 as following:
- We made sure the discussion part is in the same order as the RQs.
- We have summarized the conclusions reached along with the answer for each RQ.
- We discussed about the gaps in the literature, the points that future studies should focus on and areas with opportunity for improvement with the conclusion of each RQ.
- The remarkable points reached by the analyzes in Section 3 were mentioned in the discussion and the most important results were emphasized proportionally.
Thank you again for taking the time to share your valuable comments and suggestions with us. Please do not hesitate to write us if you have any further questions or suggestions.
Your's respectfully.
Reviewer 3 Report
The paper is an analysis of the literature, it presents an important gap, currently Bibliometric methods are used to make these types of analysis. These methods have the potential to introduce a systematic, transparent, and reproducible review process and thus improve the quality of reviews, see Aria, Cuccurullo (2017) bibliometrix: An R-tool for comprehensive science mapping analysis, Journal of informetrics 11 (4) , 959-975. Not using these methods means that the papers reported in the paper are linked to the cultural background of the authors.
However, I do not want to embarrass the authors in proposing to use such methods, but they should broaden their analysis by also introducing the following papers which include a methodology not described by the authors:
1. Ben-Akiva, M.E .; Lerman, S.R. Discrete Choice Analysis. Theory and Application to Travel Demand; MitPress, Ed .; Cambridge MA, 1985;
2. Train, K.E. Discrete Choice Methods with Simulation; Cambridge University Press: Cambridge, 2003; ISBN 9780511753930.
The predictive capability of models has been assessed, e.g. in:
3. Tinessa, F .; Papola, A .; Marzano, V. The importance of choosing appropriate random utility models in complex choice contexts. In Proceedings of the 2017 Fifth International Conference on Models and Technologies for Intelligent Transportation Systems (MT-ITS), Naples, Italy, 26-28 June 2017; pp. 884–888.
4. 1. Zhao, X .; Yan, X .; Yu, A .; Van Hentenryck, P. Prediction and behavioral analysis of travel mode choice: A comparison of machine learning and logit models. Travel Behav. Soc. 2020, 20.
5. van Cranenburgh, S .; Wang, S .; Vij, A .; Pereira, F .; Walker, J. Choice modeling in the age of machine learning - Discussion paper. J. Choice Model. 2022, 42.
Author Response
Dear reviewer,
Thank you very much for reviewing our study and sharing your valuable comments with us. We think that your suggestions are very correct and appropriate. Based on your suggestions, we made major corrections in our study.
We have examined all the papers you have suggested and added them to our study, and we have broaden our analysis and theoretical background in line with following papers:
- Aria M, Cuccurullo C. bibliometrix: An R-tool for comprehensive science mapping analysis. J Informetr. 2017;11(4):959-975. doi:10.1016/j.joi.2017.08.007
- Ben-Akiva M, Bierlaire M. Discrete Choice Methods and their Applications to Short Term Travel Decisions. In: Hall, R.W. (eds) Handbook of Transportation Science. International Series in Operations Research & Management Science. 1999;23. Springer, Boston, MA. doi.org/10.1007/978-1-4615-5203-1_2
- Train KE. Discrete Choice Methods with Simulation. Cambridge University Press: Cambridge. Vol 9780521816.; 2003. ISBN 9780511753930. doi:10.1017/CBO9780511753930
- Tinessa F, Papola A, Marzano V. The importance of choosing appropriate random utility models in complex choice contexts. 5th IEEE Int Conf Model Technol Intell Transp Syst MT-ITS 2017 - Proc. 2017;(1):884-888. doi:10.1109/MTITS.2017.8005638
- Zao X, Yan X, Yu A, Van Hentenryck P. Prediction and behavioral analysis of travel mode choice: A comparison of machine learning and logit models. Travel Behav Soc. 2020;20(February):22-35. doi:10.1016/j.tbs.2020.02.003
- van Cranenburgh S, Wang S, Vij A, Pereira F, Walker J. Choice modelling in the age of machine learning - Discussion paper. J Choice Model. 2022;42(January 2021):100340. doi:10.1016/j.jocm.2021.100340
Also for your information, we made the following corrections in line with the suggestions from reviwers:
- We added the number of studies included in our SLR to the abstract.
- We added the inclusion criteria to the Section 2.1.
- We added the range of search period to the Section 2.2.
- We majorly revised the Section 3.
- At the end of each subsection in Section 3, we've added why the answer to that RQ is important, how it will benefit to the future research, and other noteworthy highlights. We tried to mention RQs at every stage of our study. We updated the order and names of the subsections in Section 3 to have the same layout as the RQs.
- We've majorly revised and improved significantly Section 4.
- We made sure the discussion part is in the same order as the RQs.
- We have summarized the conclusions reached along with the answer for each RQ.
- We discussed about the gaps in the literature, the points that future studies should focus on and areas with opportunity for improvement with the conclusion of each RQ.
- The remarkable points reached by the analyzes in Section 3 were mentioned in the discussion and the most important results were emphasized proportionally.
Thank you again for taking the time to share your valuable comments and suggestions with us. Please do not hesitate to write us if you have any further questions or suggestions.
Your's respectfully.
Round 2
Reviewer 2 Report
The authors have improved the manuscript following all my comments. The paper can be now accepted for publication.
Reviewer 3 Report
The authors accepted all the revisions